# Glucocorticoids with low-dose anti-IL1 anakinra rescue in severe non-ICU COVID-19 infection: A cohort study

Raphael Borie[1], Laurent Savale[2], Antoine Dossier[3], Jade Ghosn[4], Camille Taillé[1], Benoit Visseaux[5], Kamel Jebreen[6], Abourahmane Diallo[6], Chloe Tesmoingt[7], Lise Morer[1], Tiphaine Goletto[1], Nathalie Faucher[8], Linda Hajouji[1], Catherine Neukirch[1], Mathilde Phillips[1], Sandrine Stelianides[1], Lila Bouadma[9], Solenn Brosseau[1], Sébastien Ottaviani[10], Johan Pluvy[1], Diane Le Pluart[4], Marie-Pierre Debray[11], Agathe Raynaud-Simon[7], Diane Descamps[5], Antoine Khalil[11], Jean Francois Timsit[9], Francois-Xavier Lescure[4], Vincent Descamps[12], Thomas Papo[3], Marc Humbert[2], Bruno Crestani[1], Philippe Dieude[10], Eric Vicaut[6☯], Gérard Zalcman[1☯*], on behalf of Bichat & Kremlin-Bicêtre AP-HP COVID teams[¶]

1 Pulmonology and Thoracic Oncology Department, University Hospital Bichat-Claude Bernard, AP-HP, Université de Paris, Paris, France, 2 Pulmonology Department, Kremlin-Bicêtre University Hospital, AP-HP, Paris-Saclay University, Kremlin-Bicêtre, France, 3 Internal Medicine Department, University Hospital Bichat-Claude Bernard, AP-HP, Université de Paris, Paris, France, 4 Infectious Disease Department, University Hospital Bichat-Claude Bernard, AP-HP, Université de Paris, Paris, France, 5 Virology Department, University Hospital Bichat-Claude Bernard, AP-HP, Université de Paris, Paris, France, 6 Biostatistics and Clinical Research Department, University Hospital Lariboisière, AP-HP, Université de Paris, Paris, France, 7 Pharmacy Department, University Hospital Bichat-Claude Bernard, AP-HP, Université de Paris, Paris, France, 8 Geriatrics Department, University Hospital Bichat-Claude Bernard, AP-HP, Université de Paris, Paris, France, 9 Medical and infectious Diseases ICU, Intensive Care Department, University Hospital Bichat-Claude Bernard, AP-HP, Université de Paris, Paris, France, 10 Rheumatology Department, University Hospital Bichat-Claude Bernard, AP-HP, Université de Paris, Paris, France, 11 Radiology Department, University Hospital Bichat-Claude Bernard, AP-HP, Université de Paris, Paris, France, 12 Dermatology Department, University Hospital Bichat-Claude Bernard, AP-HP, Université de Paris, Paris, France

☯ These authors contributed equally to this work.
¶ The complete membership of the author group can be found in the acknowledgments
* gerard.zalcman@aphp.fr

**Data Availability Statement:** Data underlying the study cannot be made publicly available due to ethical restrictions imposed by the ethical committee of Assistance Publique -Hôpitaux de

## Abstract

### Background

The optimal treatment for patients with severe coronavirus-19 disease (COVID-19) and hyper-inflammation remains debated.

### Material and methods

A cohort study was designed to evaluate whether a therapeutic algorithm using steroids with or without interleukin-1 antagonist (anakinra) could prevent death/invasive ventilation. Patients with a ≥5-day evolution since symptoms onset, with hyper-inflammation (CRP≥50mg/L), requiring 3–5 L/min oxygen, received methylprednisolone alone. Patients needing ≥6 L/min received methylprednisolone + subcutaneous anakinra daily either front-line or in case clinical deterioration upon corticosteroids alone. Death rate and death or intensive care unit (ICU) invasive ventilation rate at Day 15, with Odds Ratio (OR) and 95%

Paris (AP-HP). Data will be available on request after approval of this committee. Data requests may be sent to Comité d'évaluation de l'éthique des projets de recherche biomédicale (CEERB Paris Nord) IRB00006477 sec.ceerb@aphp.fr; or michel.lejoyeux@aphp.fr or Pr G. Zalcman, Service d'Oncologie Thoracique, Hôpital Bichat-Claude Bernard, 46 rue Henri Huchard 75018 Paris, France.

**Funding:** The authors received no specific funding for this work.

**Competing interests:** Raphael Borie, Laurent Savalle, Antoine Dossier, Camille Taillé, Benoit Visseaux, Kamel Jebreen, Sébastien Ottaviani, Chloe Tesmoingt, Lise Morer, Tiphaine Goletto, Nathalie Faucher, Linda Hajouji, Catherine Neukirch, Mathilde Phillips, Sandrine Stelianides, Solenn Brosseau, Johan Pluvy, Marie Pierre Debray, Raynaud-Simon Agathe, Antoine Khalil, Vincent Descamps, Thomas Papo, Marc Humbert, Bruno Crestani, Eric Vicaut, Gérard Zalcman have nothing to disclose. Jean Francois Timsit reported participation to an advisory board from Gilead. Is the principal investigator of PHRC-N 'Covidicus' (Dexamethasone vs. Placebo on Covid-19 pneumonia in ICUs) granted by the French Ministry of Health. Jade Ghosn reported receiving Jade Ghosn reported receiving travel grants and fundings from Gilead Sciences, ViiV Healthcare and MSD. Benoit Visseaux reported grants from QIAGEN outside the scope of the current work Xavier Lescure reported travel grants and fundings from from Gilead, MSD, Astellas, Eumedica. This does not alter our adherence to PLOS ONE policies on sharing data and materials.

CIs, were determined according to logistic regression and propensity scores. A Bayesian analysis estimated the treatment effects.

## Results

Of 108 consecutive patients, 70 patients received glucocorticoids alone. The control group comprised 63 patients receiving standard of care. In the corticosteroid±stanakinra group (n = 108), death rate was 20.4%, versus 30.2% in the controls, indicating a 30% relative decrease in death risk and a number of 10 patients to treat to avoid a death (p = 0.15). Using propensity scores a per-protocol analysis showed an OR for COVID-19-related death of 0.9 (95%CI [0.80–1.01], p = 0.067). On Bayesian analysis, the posterior probability of any mortality benefit with corticosteroids+/-anakinra was 87.5%, with a 7.8% probability of treatment-related harm. Pre-existing diabetes exacerbation occurred in 29 of 108 patients (26.9%).

## Conclusion

In COVID-19 non-ICU inpatients at the cytokine release phase, corticosteroids with or without anakinra were associated with a 30% decrease of death risk on Day 15.

## Introduction

The coronavirus disease (COVID-19) pandemic caused by the SARS-CoV-2 virus spread worldwide within 2 months [1,2]. Available data would suggest an approximately 1% global mortality and up to 15% mortality for inpatients requiring oxygen [3,4]. Antiviral therapy (remdesivir) recently showed effectiveness in a randomized controlled trial (RCT) dedicated to COVID-19-related lower respiratory tract infections. However, 7.1% of the 538 remdesivir patients had died at Day 14 [5]. Severe COVID-19 patients exhibited a so called "cytokine storm", typically 5–10 days after symptom onset [6], with fever, increased oxygen requirement, and elevated inflammatory markers [7]. While anti-inflammatory drugs such as corticosteroids or anti-interleukins (IL) therapy are still debated or contra-indicated in septic shock [8–11], such drugs may efficiently target the hyper-inflammatory phase, according to the high levels of blood cytokines at this phase of SARS-CoV-2 infection. Accordingly, the coronavirus are able to activate the NLRP3 inflammasome [12]. Although corticosteroids interfere with a wide array of inflammatory pathways, they were initially actively discouraged by WHO for COVID-19, based on small-sized studies on SARS-CoV-1 and MERS coronavirus infections [13]. In such diseases, either safety concerns or no clinical efficacy were reported [14,15], with possibly increase in viral load and prolonged viral shedding [16]. In China, however, corticosteroids were used in up to 50% of severe COVID-19 patients [17–19]. In a retrospective cohort involving 46 severe COVID-19 patients, intravenous methylprednisolone resulted in significant decreases in ICU hospitalization length, a significant improvement in SpO2, yet without a significant decrease in mortality [18]. More recently the controlled, open-label trial RECOVERY showed that oral or intravenous dexamethasone for up to ten days significantly reduced 28-days mortality by 30% as compared with "standard of care" [20].

On March 27, 2020, our multidisciplinary medical COVID-19 response team proposed a treatment algorithm for all incoming non-ICU severe patients (S1 Fig), which included

anticoagulants, corticosteroids and rescue anakinra. Results were compared with those observed in severe COVID-19 control patients, who had not received any immunosuppressive therapy.

## Methods and materials

### Study participants

All non-ICU consecutive COVID-19 inpatients that met the criteria received steroids, anakinra or both from March 27 to April 10 in Bichat University Hospital, Paris. They all exhibited positive SARS-CoV-2 RT-PCR. Patients were considered severe when meeting the following criteria: symptom duration ≥5 days; bilateral pneumonia based on non-injected thoracic CT scan, need for ≥3 L/min oxygen supply in view of ≥94% oxygen saturation measured by pulse oximetry; hyper-inflammation assessed by C-reactive protein (CRP) blood levels ≥50mg/L, or CRP between 20 and 50mg/L and increased blood ferritin (>500 μg/L) or D-dimers levels (>500 ng/mL) (S1 Fig). All patients received 120mg methyl-prednisolone (daily dose) on three consecutive days. At Day 4, if the required oxygen to ensure SpO2 ≥94% was 2 L/mn or less, intravenous corticosteroids could be switched to orally administered corticosteroids, then tapered with 40mg prednisone-equivalent for 7 days, then 20 mg for 7 days, 10 mg for another 7 days, and finally stopped. At Day 4, if ≥3 L/min was needed, 100mg anakinra daily was added subcutaneously for ≤5 days. At Day 1, if patients needed ≥6L/min oxygen flow, yet no ICU transfer/invasive ventilation, they received daily treatment of 120mg IV corticosteroids and 100mg subcutaneously anakinra for 5 days, with the same steroid tapering schedule. All patients underwent strongyloidiasis prevention on Day 1 with a single 12–15mg ivermectin dose. Thrombosis prophylaxis was given using a low-molecular-weight heparin (LMWH) dose unless medical contraindication. Patients receiving the antiretroviral lopinavir–ritonavir combination required the steroid dose to be divided by two. Patients were systematically assessed using blood cell counts, ionogram and creatininemia, liver tests, NT-pro BNP, troponin, CRP, and coagulation tests, with SARS-CoV-2 viral load assessed by real-time RT-PCR and all these data were prospectively collected. On admission, all patients underwent chest CT scans that were all reviewed by two experienced thoracic radiologists [21]. At Day 1, weight, body mass index (BMI), systolic blood pressure, and maximal body temperature were recorded, with oxygen saturation monitored from Day 1 on glucocorticoids. A retrospective cohort of severe non-ICU inpatients, hospitalized from February 15 to March 27 2020, in Bichat University Hospital (Assistance Publique-Hôpitaux de Paris [AP-HP]), before the therapeutic algorithm's setting up, and in Bicêtre University Hospital (AP-HP), from February 15 to April 18, were the historical controls. Patients from Bicêtre Hospital, though presenting with the same severity criteria did never receive corticosteroids, ivermectin, or antiviral drugs, but all received LMWH thrombosis prevention in absence of contra-indication. The date of inclusion of such patients from the retrospective cohort was comprised between 28th Feb. to 4th Apr.2020; 85% of these control patients being included (hospitalized) during March.

According to French regulatory laws, all patients whether from the prospective or the retrospective cohorts received written information by the referring physician and provided their oral consent for data collection. All patients were informed about steroids' and anakinra's off-label use. Patients' clinical charts were prospectively collected using a de-identified form. This study was approved by the local ethics committee (CEERB) of Paris Nord (Institutional Review Board-IRB 00006477, Paris-7 University, AP-HP), and by the Institutional Review Board of the French Learned Society for Respiratory Medicine (CEPRO 2020–023). Please find enclosed such approvals. No patient denied being part of the study at the date of November, 15 2020.

Patients database used for the present analyses are available on request in adherence to PLOS ONE policies on sharing data and materials.

## Virology

For all patients included in this study, diagnosis of SARS-CoV-2 infection was performed by RT-PCR on naso-pharyngeal swabs. Different techniques were performed throughout the study period, due to frequent shortage issues and requirements for fast turnaround time: Real-Star® SARS-CoV-2 (Altona, Hamburg, Germany), Cobas® SARS-CoV-2 (Roche Diagnostics, Branchburg, NJ, USA), Simplexa® COVID-19 Direct kit (DiaSorin, Gerenzano, Italy), BioFire® SARS-CoV-2 (BioMerieux, Salt Lake City, UT, USA) and QIAstat-Dx® Respiratory SARS-CoV-2 (Qiagen, Hilden, Germany).

## Statistical analysis

All consecutive patients that received at least one steroid or anakinra dose, according to the Bichat algorithm criteria, were considered. The primary endpoint was death within 15 days following COVID-19 hospitalization unit admission. The co-primary endpoint was a composite of death and invasive mechanical ventilation requirement within 15 days. The secondary endpoint was viral load in patients with iterative viral samplings. All analyses were made on the intent-to-treat population.

Group comparisons for quantitative and qualitative variables employed the t-test, Mann–Whitney test, or Chi-squared test, depending on the statistical distribution of variables.

The primary efficacy outcome was analyzed by logistic regression and propensity score methods, using a doubly robust estimator [22]. Variables *a priori* known as predictors of death were included in the regression model (i.e. age, gender, BMI, smoking, diabetes, Oxygen flow at baseline, arterial pressure and recent cancer history), were included in the propensity score model. Multiple imputation techniques were employed for missing data imputation before propensity score analysis. Sensitivity analyses using patients with no missing data on covariates were conducted to assess our conclusion's robustness. In addition, taking into account for the fact that some controls were hospitalized earlier compared to patients treated with corticosteroids, we performed an additional sensitivity analysis using the same methods but considering only controls hospitalized in the very same period as corticosteroid-treated patients. We finally performed a Bayesian propensity score analysis of our primary outcome using the same variables in propensity score building as in the frequentist approach, estimating treatment effect as Odds ratio (OR), Risk Ratio (RR) or as Absolute Risk Reduction (ARR) [23] (online statistical appendix).

In 17 patients treated with steroids and/or steroids plus ankinra, sequential viral loads were quantified by real time semi-quantitative reverse transcriptase polymerase chain reactions (RT-PCR). Results provided in Ct were transformed to $\log_{10}$ RNA copies/mL using the relationship assessed by Pasteur Institute for both genes targeted (*https://www.who.int/docs/default- source/coronaviruse/real-time-rt-pcr-assays-for-the-detection-of-sars-cov-2-institut-pasteur-paris.pdf?sfvrsn = 3662fcb6_2*). The differences between the log (number of viral copies/mL) before and after steroid treatment were tested using a paired Student's t-test.

## Results

From March 27 to April 10, 2020, 120 consecutive patients were prospectively accrued, provided they were hospitalized in a medical COVID-19 hospitalization unit (Fig 1: flow-chart). Twelve patients with solid organ transplantation (lung n = 3, kidney n = 8, heart n = 1) were excluded from the final analysis, because of their pre-existing underlying immunosuppression.

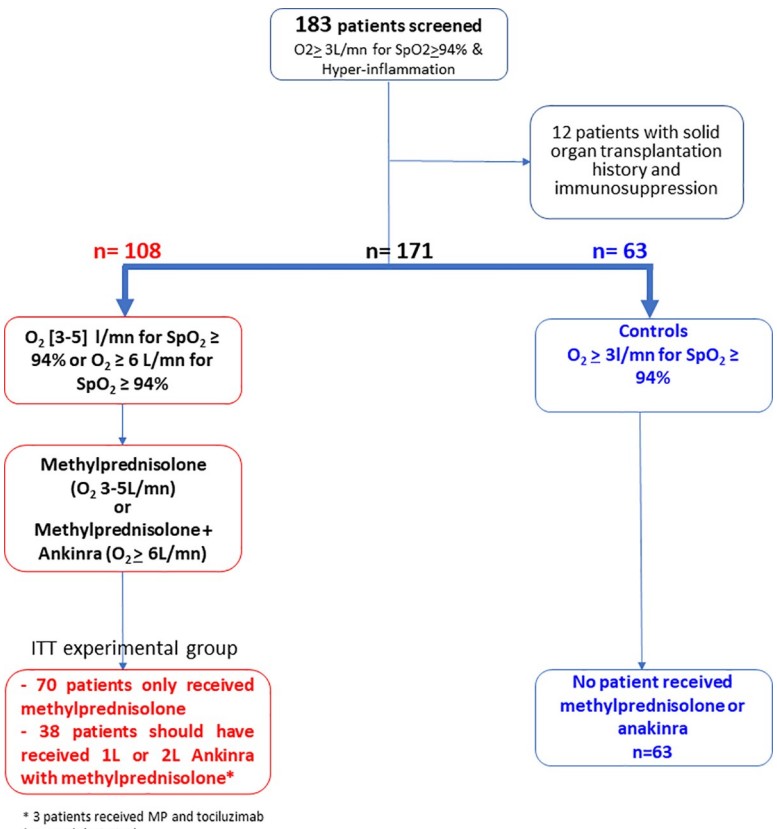

**Fig 1. Flowchart of the 120 patients included in treatment algorithm group.**

The remaining patients received corticosteroids alone (n = 70), frontline corticosteroids plus anakinra (n = 12), second-line anakinra following clinical deterioration or no improvement at Day 4 of steroids (n = 26) (Fig 1). Instead of anakinra, three patients received tocilizumab at Day 3 or 4 following corticosteroid start, due to clinical deterioration, which was considered as a protocol deviation. These patients were however included in the intent-to-treat group. The final study population comprised 108 patients (Fig 1), and the control group 63 patients (Bichat hospital: n = 21, Kremlin-Bicêtre hospital: n = 42). Tables 1–3 list the patients' baseline characteristics. 18 patients were transferred to ICU for invasive ventilation within the 15 days post steroids initiation.

Both groups only differed as to a higher number of cancer patients (p = 0.03) in the cortico-steroid group and higher blood pressure in the control group (p = 0.0003.). Although the control group displayed a slightly higher D-dimer concentration (p = 0.04), the incidence of venous thrombosis or pulmonary embolism, did not significantly differ between both groups. Respectively 8 (7.4%) and 3 (4.8%) patients in the steroid and control groups respectively (p = 0.75), had thromboembolic disease diagnosed during the first 15 days of hospitalization and received LMWH at curative, anticoagulant dosing. The two groups did not significantly differ according to CRP, ferritin and fibrinogen blood levels, suggesting they did not differ according to the overall level of systemic inflammation. Control group patients received the lopinavir–ritonavir combination, hydroxychloroquine, and ivermectin significantly less frequently (S1 Table). Three patients received remdesivir, while 89% and 95% of patients received thrombosis prophylaxis with LMWH in the steroid and control groups, respectively.

**Table 1. Baseline (D1 of steroids, for steroid group) clinical characteristics in overall population, in the steroids and the control group.**

| | Overall population (n = 171) | Steroids-based treatment group (n = 108) | Control group (n = 63) | p-value |
|---|---|---|---|---|
| **Age** at diagnosis median (IQR) | 67.1 (56.7–78.1) | 67. 9 (56.7–77.7) | 66.3 (55.9–78.1) | 0.88* |
| **Gender**: | | | | |
| Female, n (%) | 48 (28.1%) | 27 (25.0%) | 21 (33.3%) | 0.24£ |
| Male, n (%) | 123 (71.9%) | 81 (75.0%) | 42 (66.7%) | |
| **Active smoking**-no. (%) | | | | |
| Yes | 10/160 (6.3%) | 6/104 (5.8%) | 4/56 (7.1%) | 0.74¶ |
| No | 150/160 (93.8%) | 98/104 (94.2%) | 52/56 (92.9%) | |
| **Pack-years** | | | | |
| n (miss) | 63(108) | 36(72) | 27(36) | 0.0002† |
| Mean ± sd | 24.5 ± 25.2 | 30.6 ± 18.4 | 16.4 ± 30.7 | |
| **Weight** | | | | |
| n (miss) | 146(25) | 97(11) | 49(14) | 0.48† |
| median (IQR) | 80.0 (70.0–92.0) | 79.0 (69.0–92.0) | 80.0 (71.0–91.0) | |
| **BMI** | | | | |
| n (miss) | 131(40) | 89(19) | 42(21) | 0.18† |
| median (IQR) | 26.9 (24.4–31.1) | 26.8 (24.0–30.4) | 27.9 (25.2–31.2) | |
| **duration of symptoms** | | | | |
| n (miss) | 169(2) | 107(1) | 62(1) | 0.99† |
| median (IQR) | 7.0 ± 4.3 | 7.0 ± 4.5 | 6.9 ± 4.0 | |
| **duration of fever** | | | | |
| n (miss) | 159(12) | 101(7) | 58(5) | 0.50† |
| median (IQR) | 4.1 ± 4.3 | 3.8 ± 4.0 | 4.6 ± 4.8 | |
| **Max. Temp. at D1** (°C) | | | | |
| n (miss) | 164(7) | 104(4) | 60(3) | 0.21† |
| median (IQR) | 38.6 (38.0–39.2) | 38.6 (37.9–39.1) | 38.7 (38.3–39.3) | |
| **SpO2** (%) median (IQR) | 95.0 (93.0–96.0) | 95.0 (93.0–96.0) | 94.0 (93.0–95.0) | 0.01† |
| **O2 flow** median (IQR) | 5.0 (4.0–8.0) | 5.0 (4.0–9.0) | 5.0 (4.0–6.0) | 0.78† |
| **Systolic blood pressure** (mmHg) median (IQR) | 124.0 (111.0–140.0) | 119.6 (109.0–138.0) | 129.0 (122.0–150.0) | 0.0004* |
| **Diastolic blood pressure** (mmHg) median (IQR) | 71.0 (62.0–82.0) | 70.0 (60.0–80.0) | 75.0 (63.0–87.0) | 0.045† |

* Two-Sample T-test

† Mann Whitney U test/Wilcoxon Sum Rank test

¶ Fisher's exact test

£ Pearson's chi-square test. BMI body mass index.

Considering both co-primary endpoints, there was a 33.3% death or ICU transfer rate at Day 15 in the 171-patient population (n = 57) and 24.0% rate of COVID-19-related deaths (n = 41). In the cortcicosteroid group, a 29.6% death or ICU transfer rate was noted versus 39.7% in the controls (p = 0.18). In the steroid group, there was an observed death rate of 20.4%, versus 30.2% in the controls (p = 0.15). Kaplan Meier curves of the time to death from day one of hospitalization are shown in Fig 2.

The association between clinical and biological variables and the death risk is shown in Table 4. Patients that died were significantly older (median age difference = 18 years) than

**Table 2. Baseline (D1 of steroids, for steroid group) biological data in overall population and in the steroids and the control group.**

| | Overall population (n = 171) | Steroids-based treatment group (n = 108) | Control group (n = 63) | p-value |
|---|---|---|---|---|
| **Leucocytes (G/L)** Median (IQR) | 6.8 (5.5–8.6) | 6.9 (5.4–8.6) | 6.6 (5.6–8.9) | 1.00[†] |
| **Neutrophils (G/L)** Median (IQR) | 5.30 (4.29–7.26) | 5.24 (4.28–7.00) | 5.36 (4.47–7.34) | 0.56[†] |
| **Lymphocytes (G/L)** | | | | |
| n (miss.) | 169(2) | 108(0) | 61(2) | 0.54[†] |
| Median (IQR) | 0.81 (0.59–1.26) | 0.81 (0.59–1.21) | 0.86 (0.61–1.38) | |
| **Monocytes (G/L)** | | | | |
| n (miss.) | 168(3) | 108(0) | 60(3) | 0.46[†] |
| Median (IQR) | 0.40 (0.25–0.57) | 0.41 (0.24–0.59) | 0.37 (0.27–0.53) | |
| **Platelets (G/L)** | | | | |
| n (miss.) | 168(3) | 107(1) | 61(2) | 0.48[†] |
| Median (IQR) | 199 (158–273) | 199 (158–280) | 196 (157–259) | |
| **Hb (g/L)** | | | | |
| n (miss.) | 170(1) | 108(0) | 62(1) | 0.29[†] |
| median (IQR) | 12.7 (11.7–13.9) | 12.7 (11.7–13.9) | 12.9 (11.9–13.9) | |
| **Total Bilirubin (µmol/L)** | | | | |
| n (miss.) | 151(20) | 97(11) | 54(9) | 0.49[†] |
| median (IQR) | 9 (6–12) | 9 (6–12) | 9 (7–12) | |
| **ASAT (UI/L)** | | | | |
| n (miss.) | 155(16) | 100(8) | 55(8) | 0.44[†] |
| median (IQR) | 51 (36–76) | 51 (35–75) | 54 (38–77) | |
| **ALAT (UI/L)** | | | | |
| n (miss.) | 155(16) | 100(8) | 55(8) | 0.26[†] |
| median (IQR) | 35 (26–56) | 35 (27–56) | 34 (23–56) | |
| **Albumine (g/L)** | | | | |
| n (miss) | 84(87) | 67(41) | 17(46) | 0.74[†] |
| median (IQR) | 30 (27–32 | 30 (26–32) | 30 (27–35) | |
| **Urea (nmol/L)** | | | | |
| n (miss) | 167(4) | 106(2) | 61(2) | 0.25[†] |
| median (IQR) | 6.4 (4.5–10.0) | 7.0 (4.9–10.7) | 5.5 (4.5–8.9) | |
| **Creatininemia (µmol/L)** | | | | |
| n (miss) | 168(3) | 107(1) | 61(2) | 0.75[†] |
| median (IQR) | 80 (62–113) | 82 (62–114) | 79 (62–108) | |
| **CPK (U/L)** | | | | |
| n (miss) | 125(46) | 75(33) | 50(13) | 0.69[†] |
| median (IQR) | 159 (70–357) | 158 (68–385) | 174 (70–289) | |
| **LDH (U/L)** | | | | |
| n (miss) | 122(49) | 76(32) | 46(17) | 0.03[†] |
| median (IQR) | 423 (338–500) | 401 (325–481) | 440 (367–531) | |
| **NT-proBNP (ng/L)** | | | | |
| n (miss) | 102(69) | 77(31) | 25(38) | 0.34[†] |
| median (IQR) | 325 (94–1097) | 2715.0 ± 9397.8 | 383 (92–2080) | |
| **CRP (mg/mL)** | | | | |
| n (miss) | 163(8) | 105(3) | 58(5) | 0.08[†] |
| median (IQR) | 141 (99–194) | 134 (91–183) | 148 (105–211) | |
| **Ferritin (µg/L)** | | | | |

*(Continued)*

**Table 2.** (Continued)

| | Overall population (n = 171) | Steroids-based treatment group (n = 108) | Control group (n = 63) | p-value |
|---|---|---|---|---|
| n (miss) | 44(127) | 38(70) | 6(57) | 0.11[†] |
| median (IQR) | 1047 (521–1945) | 962 (489–1883) | 1945 (1863–3480) | |
| **Fibrinogen (g/L)** | | | | |
| n (miss) | 108(63) | 63(45) | 45(18) | 0.3[*] |
| median (IQR) | 6.13 (5.11–6.98) | 6.13 (5.11–6.66) | 6.30 (5.20–7.40) | |
| **D-dimers (ng/mL)** | | | | |
| n (miss) | 90(81) | 45(63) | 45(18) | 0.044[†] |
| median (IQR) | 1073 (680–2051) | 946 (629–1399) | 1410 (820–2370) | |

[*] Two-Sample T-test

[†] Mann Whitney U test/Wilcoxon Sum Rank test

[¶] Fisher's exact test

[£] Pearson's chi-square test.

Abbreviations: D, day; Hb hemoglobin, ASAT, aspartate aminotransferase; ALAT, alanine aminotransferase; CPK, creatine phosphokinase; LDH, lactate dehydrogenase; NT-proBNP, N-terminal pro-brain natriuretic peptide; CRP, C reactive protein; IQR, interquartile range; miss, missing.

those that survived (p <0.0001). They suffered more often from hypertension (p = 0.012) and diabetes (p = 0.0001), and presented higher plasma CRP and D-dimer levels (p = 0.002 and 0.017, respectively). Notably, male gender, active smoking and higher BMI were not associated with a significantly higher death risk (Table 4).

Using propensity scores with the "double robust method" (22), the OR for death or ICU transfer was 0.88 (95% IC [0.77–1.02], p = 0.08), and the OR for COVID-19-related death was 0.91 (95% IC [0.81–1.01], p = 0.09). A sensitivity analysis, which considered the three patients that received tocilizumab instead of anakinra and those that received remdesivir instead of lopinavir/ritonavir, reinforced such trend with an OR for death or ICU transfer of 0.88 (95% IC [0.76–1.02], p = 0.088) and an OR for COVID-19-related death of 0.90 (95% IC [0.8–1.01], p = 0.067). An additional sensitivity analysis on the patients with no missing data on covariates (N = 151) found the OR for death or ICU transfer equal to 0.84 (95% IC [0.72–0.99], p = 0.02), and the OR for COVID-19-related death equal to 0.87 (95% IC [0.74–1.00], p = 0.06). Finally, a time-restricted analysis to the period of time from 27th March to 10th April only including 42 control patients along with the 108 patients of the experimental group, showed similar COVID19-related death rate at D15 of 31.0% (13/42) in the control strictly contemporary group. OR for COVID19-related death was 0.86 (0.72–1.02), p = 0.07, and OR for COVID19-related death/ICU transfer 0.86 (0.71–1.04), p = 0.11, while in the subset of patients without any missing covariates during the very same period, the OR for COVID19-related death was 0.80 (0.59–1.02), p = 0.07, the OR for COVID19-related death/ICU transfer 0.80 (0.61–1.01), p = 0.056.

In a Bayesian analysis that used non-informative priors, the estimates adjusted with propensity scores and their 95% credible intervals (CrIs) were 0.63 (95% CrI, 0.27–1.24) for OR, 0.72 (95% CrI, 0.37–1.31) for RR, and −0.10 (−0.24, 0.04) for absolute risk reduction (ARR). The posterior probability of any mortality benefit with steroids (i.e., RR <1) was 87.5%, and the probability of an RR of <0.9 was 80.7%. Assuming a 30% death risk in controls, the probability of an ARR of ≥2% was 84%, and ARR of ≥10% was 48% (S2 Table). The posterior probability of any treatment-related harm, defined as an OR >1, was 7.8%. The S2 Fig. presents the posterior probability distribution for RR reductions for non-informative priors. Results obtained when considering enthusiastic or skeptical priors are shown in the S2 Table.

**Table 3. Comorbidities in overall population and in the steroids and the control group.**

| | Overall population (n = 171) | Steroids-based treatment group (n = 108) | Control group (n = 63) | p-value |
|---|---|---|---|---|
| **COPD** | | | | |
| Yes | 13 (7.6%) | 7 (6.5%) | 6 (9.5%) | 0.55[¶] |
| No | 158 (92.4%) | 101 (93.5%) | 57 (90.5%) | |
| **Hypertension** | | | | |
| Yes | 92 (53.8%) | 59 (54.6%) | 33 (52.4%) | 0.78[£] |
| No | 79 (46.2%) | 49 (45.4%) | 30 (47.6%) | |
| **Diabetes** | | | | |
| Yes | 54 (31.6%) | 29 (26.9%) | 25 (39.7%) | 0.08[£] |
| No | 117 (68.4%) | 79 (73.1%) | 38 (60.3%) | |
| **Ischemic cardiopathy** | | | | |
| Yes | 26 (15.2%) | 15 (13.9%) | 11 (17.5%) | 0.53[£] |
| No | 145 (84.8%) | 93 (86.1%) | 52 (82.5%) | |
| **Active cancer <18 mo** | | | | |
| Yes | 13 (7.6%) | 12 (11.1%) | 1 (1.6%) | 0.65[¶] |
| No | 158 (92.4%) | 96 (88.9%) | 62 (98.4%) | |
| **Lung fibrosis** | | | | |
| Yes | 8 (4.7%) | 5 (4.6%) | 3 (4.8%) | 1.00[¶] |
| No | 163 (95.3%) | 103 (95.4%) | 60 (95.2%) | |
| **Cardiac insufficiency** | | | | |
| Yes | 19 (11.1%) | 13 (12.0%) | 6 (9.5%) | 0.614[£] |
| No | 152 (88.9%) | 95 (88.0%) | 57 (90.5%) | |
| **Peripheral artery disease** | | | | |
| Yes | 9 (5.3%) | 7 (6.5%) | 2 (3.2%) | 0.49[¶] |
| No | 162 (94.7%) | 101 (93.5%) | 61 (96.8%) | |
| **Brain stroke history** | | | | |
| Yes | 20 (11.7%) | 14 (13.0%) | 6 (9.5%) | 0.50[£] |
| No | 151 (88.3%) | 94 (87.0%) | 57 (90.5%) | |
| **Chronic renal failure** | | | | |
| Yes | 28 (16.4%) | 21 (19.4%) | 7 (11.1%) | 0.16[£] |
| No | 143 (83.6%) | 87 (80.6%) | 56 (88.9%) | |
| **HIV positive** | | | | |
| Yes | 4 (2.3%) | 3 (2.8%) | 1 (1.6%) | 1.00[¶] |
| No | 167 (97.7%) | 105 (97.2%) | 62 (98.4%) | |
| **Cirrhosis** | | | | |
| Yes | 7 (4.1%) | 4 (3.7%) | 3 (4.8%) | 0.71[¶] |
| No | 164 (95.9%) | 104 (96.3%) | 60 (95.2%) | |
| **B/C Chronic viral hepatitis** | | | | |
| Yes | 5 (2.9%) | 4 (3.7%) | 1 (1.6%) | 0.65[¶] |
| No | 166 (97.1%) | 104 (96.3%) | 62 (98.4%) | |

[*] Two-Sample T-test

[†] Mann Whitney U test/Wilcoxon Sum Rank test

[¶] Fisher's exact test

[£] Pearson's chi-square test.

Abbreviations: COPD, chronic obstructive pulmonary disease; HIV, human immunodeficiency virus.

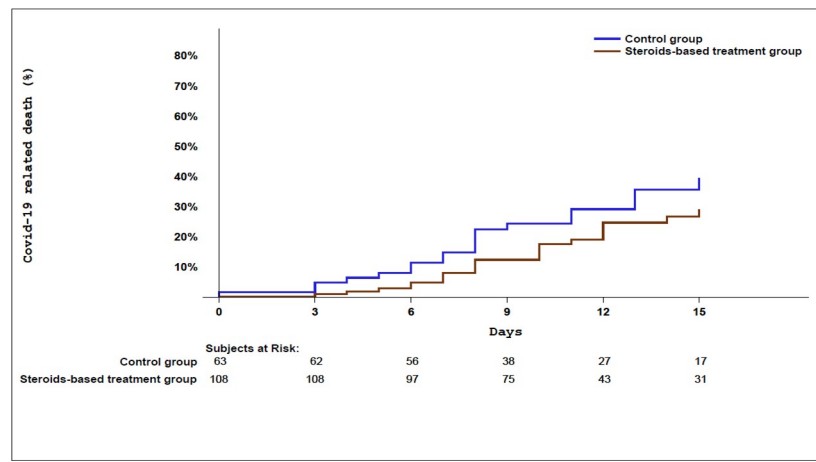

**Fig 2. Kaplan Meier curves of survival of time to death from day of hospitalization.**

**Table 4. Characteristics of the patients according to status at D15.**

| | Overall population (n = 171) | Patients deceased (n = 41) | Patients alive at D15 (n = 130) | p-value |
|---|---|---|---|---|
| **Age** at diagnosis median (IQR) | 67.1 (56.7–78.1) | 79.9 (74.2–85.3) | 61.6 (54.6–72.8) | < .0001* |
| **Gender**: | | | | |
| Female, n (%) | 48 (28.1%) | 8 (19.5%) | 40 (30.7%) | 0.16£ |
| Male, n (%) | 123 (71.9%) | 33 (80.4%) | 90 (69.2%) | |
| **Active smoking**-no. (%) | | | | |
| Yes | 10/160 (6.3%) | 4/35 (11.4%) | 6/125 (4.8%) | 0.22‡ |
| No | 150/160 (93.8%) | 31/35 (88.5%) | 119/125 (95.2%) | |
| missing data (n) | 11 | 6 | 5 | |
| **Hypertension** | | | | |
| Yes | 92 (53.8%) | 29 (70.7%) | 63 (48.4%) | 0.012£ |
| No | 79 (46.2%) | 12 (29.2%) | 67 (51.5%) | |
| **Diabetes** | | | | |
| Yes | 54 (31.6%) | 23 (56.0%) | 31 (23.8%) | 0.0001£ |
| No | 117 (68.4%) | 18 (43.9%) | 99 (76.1%) | |
| **BMI** | | | | |
| n (miss) | 131(40) | 27 (14) | 104(26) | 0.95† |
| median (IQR) | 26.9 (24.4–31.1) | 26.8 (24.7–31.6) | 26.9 (24.3–30.8) | |
| **CRP (mg/mL)** | | | | |
| n (miss) | 163(8) | 41(0) | 122(8) | 0.002† |
| median (IQR) | 141 (99–194) | 185 (108–264) | 134 (91–172) | |
| **D-dimers (ng/mL)** | | | | |
| n (miss) | 90(81) | 22(19) | 68(62) | 0.017† |
| median (IQR) | 1073 (680–2051) | 1464 (880–2246) | 952 (524–1628) | |
| **O2 flow D1 Steroids (L/min)** Median (IQR) | 5.0 (4.0–8.0) | 9.0 (4.0–12.0) | 5.0 (4.0–6.0) | 0.0007† |

* Two-Sample T-test.

† Mann Whitney U test/Wilcoxon Sum Rank test.

‡ Fisher's exact test.

£ Pearson's chi-square test.

Abbreviations: D, day; BMI, body mass index; IQR, interquartile range; CRP, C reactive protein; miss, missing.

Short-term toxicity within the first 15 days was not unexpected, essentially related to pre-existing type 2 diabetes exacerbations in 29 of 108 patients (26.9%), with oral anti-diabetic drugs alone required in 10 patients (34.5%), and insulin therapy in 19 patients (65.5%). Only three patients with steroid-induced diabetes exacerbation died, all from respiratory distress, while diabetes was correctly controlled with insulin. Pseudomonas aeruginosa pneumonia and invasive aspergillosis occurred simultaneously in one patient following ICU admission. A case of abdominal varicella-zoster occurred in one patient, with favorable evolution upon valaciclovir.

Repeated rhino-pharyngeal swabs or bronchial samplings were performed at the physician discretion. Only 17 of 108 patients in the steroid group underwent repeated SARS-CoV-2 RT-PCR on at least two samplings (2–7, mean 3.5). In these 17 patients, one of the samples was analyzed 12 hours to 6 days (mean 2.4 days) before steroids' initiation, while the second sample was analyzed at least 24 hours (1–22 days, mean 9.8) after the first dose of steroids. Among these 17 patients, 10 (58.8%) showed a decreased viral load at the first repeated sampling, with three patients having undetectable SARS-CoV-2. Conversely, six showed a stable viral load defined by a variation in the log number of viral copies by mL of $< 2$ log. Only one patient displayed a slight increase in viral load (by 2.2 log). Overall, SARS-CoV-2 became undetectable in nine patients's (52.9%) respiratory samples, with a median time of 11.5 $+/-$ 10.9 days. The viral load values, before and at least 24 hours following steroid treatment initiation, revealed a significant viral load decrease ($p = 0.008$, Student paired t-test).

## Discussion

By using short-term immunosuppression in severe COVID-19 patients, we observed a lower death rate (22.2%) in 108 patients treated with corticosteroids versus 63 patients treated without glucocorticoids (33.3%). Although COVID-19 is mostly a benign disease [24,25], some patients develop an excessive release of pro-inflammatory cytokines from Day 5–10 after symptoms onset, which is referred to as cytokine storm [26]. Such a cytokine storm has been associated with bilateral pneumonia and increased oxygen needs, rapidly evolving into acute respiratory distress syndrome (ARDS). Despite invasive ventilation, its fatality rate is high: 28%–60% [27,28]. COVID-19-related cytokine storm is reminiscent of immune-mediated diseases, such as hemophagocytic lymphohistiocytosis, cytokine release in leukemia patients receiving engineered T-cell therapy [29,30], and undesirable side effects of immune checkpoint inhibitors [31]. Owing to their mitigated outcome in the MERS-CoV [13,14], SARS-CoV-1 infections [16], and ARDS [32], the WHO did not recommend corticosteroid use. Given that corticosteroids are cheap and widely available, we felt they should be tested in COVID-19-related hyper-inflammation. In our cohort, we applied a treatment algorithm in severe COVID-19 patients selected based on rigorous criteria including symptom duration of $\geq$5days, respiratory insufficiency and systemic hyper-inflammation. The high death risk, previously reported in such patients [28], was confirmed in our series: over one-fifth of steroid group patients and one-third of control group patients died within 15 days.

While both groups were well balanced concerning clinical and biological variables, they differed in co-medication exposure (S1 Table), with less lopinavir/ritonavir, hydroxychloroquine, and ivermectin exposure in the controls. As the first two drugs failed to modify death risk in COVID-19 patients [33,34], and since ivermectin's antiviral effect was only reported *in vitro* [35], the probabilities that these differences could have influenced survival are low. Our study confirmed previously reported prognosticators, including age, male gender, hypertension, diabetes, active smoking, increased CRP, and elevated D-dimer levels (Table 4). A substantially lower death rate was observed in patients receiving corticosteroids or anakinra versus controls,

though this clinically meaningful 11% crude decrease in death rates was not statistically significant. A sensitivity analysis, which considered three patients that received tocilizumab instead of anakinra and three others that received remdesivir instead of lopinavir/ritonavir, increased such trends. These results align with those of a recent report involving 213 COVID-19 patients, 81 of whom did not receive corticosteroids, while 132 did receive corticosteroids [36]. However, our 2-week treatment duration was short, in an effort to limit side effects and long-term risk of lung fibrosis remains a concern.

No increased viral shedding occurred in the 17-patient subset that received glucocorticoids and underwent repeated RT-PCR on respiratory samples after steroids initiation. These findings reinforce those of the Lee et al study, involving 16 SARS-CoV-1 infected-patients [16], who showed that only early corticosteroid administration during viral replication, meaning the first illness days, resulted in delayed viral clearance and higher plasma viral load.

Only 35 patients received anakinra combined with corticosteroids. We are thus unable to assess anakinra's specific contribution to the therapeutic algorithm, which also includes thrombosis prevention. Other groups reported beneficial anakinra effects in severe COVID-19 patients [37–39] at higher anakinra dosing, leading to infectious complications [39,40]. In all these studies, some patients were allowed to receive corticosteroids, and the respective contribution of each treatment remains therefore unclear either. A time-restricted analysis only including patients treated at the very same period of time, over 2 weeks and half, excluded any influence of the time on our results, that the results could have improved over time because of general improvements in patient management or reduced severity of COVID-19 cases, or that a change in ICU indications or accessibility could have occurred over time. Lastly, our Bayesian analysis allows an alternative approach regarding practical conclusions of our study by showing that there is a high probability (87.5%) that the tested strategy has benefits, and a 81% probability that it could reduce risk of death by 10% that are certainly meaningful information to decide about treatment strategy in the context of pandemic.

Our study clearly displays several limitations, particularly as to its non-randomized design, leading to subtle differences in the two groups (such as hypertension frequency), and the non-blinded nature of the intervention. While being a real-life study, it involved only two centers. Another caveat is that treatment related immunosuppression could favor secondary infections. Notably, all discharged patients benefited from a web-based follow-up (COVIDOM: *https://www.nouveal.com/covidom-le-suivi-des-patientsporteurs-du-covid-19/*) [41] and phone call at home, and no such long-term adverse event was detected.

The therapeutic algorithm established and validated by the consortium of MDs in charge of COVID-19 patients in Medicine departments (ICU department were not included since ICU transfer was an endpoint of the study) was distributed to all prescribing doctors (including titular doctors or junior doctors i.e. residents) of the hospital and the information on the prospective study provided to the more than 80 physicians involved. So all patients with the *bona fide* inclusion criteria were accrued in this observational study.

However, in addition to the recently published results of the randomized RECOVERY trial that confirm corticosteroids role in severe COVID-19 patients [20], our data deserve a dedicated randomized trial evaluating the algorithm with corticosteroids and rescue anakinra, in severe COVID-19.

## Supporting information

**S1 Fig. Study design.**
(TIF)

**S2 Fig. Posterior probabilities of Relative Risk (RR) of corticosteroids administration considering a non informative prior and using Bayesian propensity score approach.**
(TIF)

**S1 Table. Co-Medications in the overall population and in the steroid and control groups.**
(DOCX)

**S2 Table. Posterior probabilities of the treatment benefit.**
(DOCX)

**S1 File.**
(DOCX)

## Acknowledgments

The authors thank all medical and paramedical teams from Hospital Bichat-Claude Bernard and Kremlin-Bicêtre hospitals who faced COVID-19 epidemic with dedication and competence, with a special thanks to Céline Namour and Zohra Brouk, the clinical research assistants of the Thoracic Oncology Unit, who assured expert data-management of the study. We thank the cardiologists Dr. Olivier Milleron and Dr. Gregory Ducrocq, for their precious help in COVID-19 patients care.

The authors would also like to express their gratitude to all the residents who faced this epidemic with courage and dedication: Mayda Al Rahi, Julien Bermudez, Thomas Bernard, Timothée Bironne, Agathe Bounhiol, Charlotte Casadepax, Jeanne Chauffier, Céline Cheron, Jonathan Cortese, William Danjou, Clémence David, Chloé De Broucker, Arthur Delayre, Julien Dessajan, Caroline Diou, Marc Doman, Dora Dreyfuss, Alexandre Egea, Valentine Ferre, Flora Finet, Quentin Fossé, Madeleine Franc, Matthieu Gabenesch, Lucile Garrault, Simon Gressens, François Grolleau, Nadia Guezour, Pierre Maya Husain, Antoine Juge, Thomas Lacoste, Terence Langlois, Ibtissem Laouati, Julie Larue, Arnaud Le Flécher, François Maillet, Clarisse Marcombes, Victor Mardon, Hugo Martiniere, Justine Mirete, Hugo Moisset, Alexandra Mokrzycki, Julie Molle, Quentin Moyon, Anne Murarasu, Clément Nachef, Sophie Nagle, Andrei Neagu, Héloïse Paugoy, Clara Pouchelon, Hélène Pringuez, Amélie Recoing, Violette Regnault De Savigny, Mathilde Salpin, Amre Shalaby, Salome Schlupmann, Sabina Solinas, Bérénice Souhail, Jihane Souilamas, Hassan Tharin, Charlotte Thibault de Menonville, Volpe Thomas, Pierre Thoré, Mickael Thy, Chloé Tridon, Simon Valayer, Charles Vauchier,

## Author Contributions

**Conceptualization:** Francois-Xavier Lescure, Vincent Descamps, Bruno Crestani, Philippe Dieude, Gérard Zalcman.

**Data curation:** Raphael Borie, Laurent Savale, Antoine Dossier, Jade Ghosn, Camille Taillé, Benoit Visseaux, Chloe Tesmoingt, Lise Morer, Tiphaine Goletto, Nathalie Faucher, Linda Hajouji, Catherine Neukirch, Mathilde Phillips, Sandrine Stelianides, Lila Bouadma, Solenn Brosseau, Sébastien Ottaviani, Johan Pluvy, Diane Le Pluart, Marie-Pierre Debray, Agathe Raynaud-Simon, Diane Descamps, Antoine Khalil, Jean Francois Timsit, Francois-Xavier Lescure, Vincent Descamps, Thomas Papo, Marc Humbert, Bruno Crestani, Philippe Dieude, Eric Vicaut, Gérard Zalcman.

**Formal analysis:** Kamel Jebreen, Abourahmane Diallo, Gérard Zalcman.

**Methodology:** Chloe Tesmoingt.

**Writing – original draft:** Philippe Dieude, Gérard Zalcman.

**Writing – review & editing:** Gérard Zalcman.

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
