## [Decision Letter · Decision Letter 0]

13 Nov 2020

PONE-D-20-25884

Glucocorticoids with low-dose Anti-IL1 Anakinra Rescue in Severe Non-ICU COVID-19 Infection: a Cohort Study

PLOS ONE

Dear Dr. Borie,

Thank you for submitting your manuscript to PLOS ONE. After careful consideration, we feel that it has merit but does not fully meet PLOS ONE’s publication criteria as it currently stands. Therefore, we invite you to submit a revised version of the manuscript that addresses the points raised during the review process.

The topic is interesting. HOwever, major issues showed be raised. The Authors should better clarified the inclusion (and exclusion) criteria for each treatment and discuss the results accordingly.  

We look forward to receiving your revised manuscript.

Kind regards,

Chiara Lazzeri

Academic Editor

PLOS ONE

Journal Requirements:

2. Please provide additional details regarding participant consent. In the ethics statement in the Methods and online submission information, please clarify whether (1) consent was informed and (2) how oral consent was documented and witnessed. If your study included minors, state whether you obtained consent from parents or guardians. Also, please clarify whether ethics approval and informed consent was obtained from patients in the prospective and retrospective cohorts.

3. In your Methods section, please provide additional information about the participant recruitment method for the prospective part of your study and the demographic details of your participants. Please ensure you have provided sufficient details to replicate the analyses such as:

a) a statement as to whether your sample can be considered representative of a larger population, and

b) a description of how participants were recruited.

4. For the retrospective data collected in your study, please include the date(s) on which you accessed the records to obtain the data used in your study.

5. Please provide the name and catalog number of the RT-PCR kit used.

6. Thank you for stating the following in the Competing Interests section:

"Raphael Borie, Laurent Savalle, Antoine Dossier, Camille Taillé, Benoit Visseaux, Kamel Jebreen, Sébastien Ottaviani, Chloe Tesmoingt,  Lise Morer, Tiphaine Goletto, Nathalie Faucher, Linda Hajouji, Catherine Neukirch, Mathilde Phillips, Sandrine Stelianides, Solenn Brosseau, Johan Pluvy, Marie Pierre Debray, Raynaud-Simon Agathe, Antoine Khalil, Vincent Descamps, Thomas Papo, Marc Humbert, Bruno Crestani, Eric Vicaut, Gérard Zalcman have nothing to disclose.

Jean Francois Timsit reported participation to an advisory board from Gilead. Is the principal investigator of PHRC-N 'Covidicus' (Dexamethasone vs. Placebo on Covid-19 pneumonia in ICUs) granted by the French Ministry of Health.

Jade Ghosn reported receiving Jade Ghosn reported receiving travel grants and fundings from Gilead Sciences, ViiV Healthcare and MSD.

Benoit Visseaux reported grants from QIAGEN outside the scope of the current work

Xavier Lescure reported travel grants and fundings from from Gilead, MSD, Astellas, Eumedica."

7. One of the noted authors is a group or consortium (Bichat & Kremlin-Bicêtre AP-HP COVID teams). In addition to naming the author group, please list the individual authors and affiliations within this group in the acknowledgments section of your manuscript. Please also indicate clearly a lead author for this group along with a contact email address.

Reviewers' comments:

Reviewer's Responses to Questions

**Comments to the Author**

1. Is the manuscript technically sound, and do the data support the conclusions?

Reviewer #1: Partly

2. Has the statistical analysis been performed appropriately and rigorously? 

Reviewer #1: Yes

3. Have the authors made all data underlying the findings in their manuscript fully available?

Reviewer #1: No

4. Is the manuscript presented in an intelligible fashion and written in standard English?

Reviewer #1: Yes

5. Review Comments to the Author

Reviewer #1: The manuscript by Raphael Borie and Colleagues explore an interesting and up to date field regarding the possible immunosuppressive therapies in course of the inflammatory phase of Covid-19, but some concerns needs to be clarified.

Major points

- The corticosteroid group is heterogeneous, including also patients receiving not only steroids, but also anakinra (35/108). The authors assess that anakinra group is too small in order to drive any conclusion about the effectiveness of this drug. In any case it is opinion of the reviewer that some information about this biologic treatment should be shown: at least its effect regard PCR, Ferritin, Fibrinogen, D-dimer concentration, PaO2/FiO2 ratio …. Otherwise anakinra group should be removed.

- Moreover the authors should report any possible side effect, or absence of side effects of anakinra treatment compared to steroids alone.

- According to the inclusion criteria for anakinra therapy, only patients requiring ≥ 6L /min O2 therapy from day one, or ≥ 3L /min O2 therapy from day 4, received anakinra (together with steroids), while those patients that were in better clinical condition received steroids alone. It is difficult to expect a better outcome for patients starting from a worst clinical condition, independently from the therapy regimen. If no differences raised from the comparison of the two groups, then the author can drive the conclusion that anakinra does not get worst the prognosis. This possible result implicitly suggests some possible positive effect of anakinra, being the starting population in a more severe condition. In any case, as the authors assess, a controlled randomised study is needed.

- What are the result of the comparison between the control group (n 63) respect to steroids only group (n 70)?

- The control group received less frequently standard therapy as compared to corticosteroid ± anakinra group. This could be a bias of the study, even if the probability that some of these drugs could significantly modify the natural history of the disease is low, as the authors assert in the discussion. The statistical analysis should be reconsidered after removing, from the control group, those patients not treated with the same standard therapy as the corticosteroid ± anakinra group.

Minor points

- Abstract, line 12: it is not clear if the authors refer to the “corticosteroids + anakinra group” or to the “corticosteroids + anakinra, plus the corticosteroids alone group”. The sentence “corticosteroids ± anakinra group” should be used.

- Discussion, page 13, line 13: “anakinra or steroids” should be probably changed with “corticosteroid ± anakinra”

6. PLOS authors have the option to publish the peer review history of their article (what does this mean?). If published, this will include your full peer review and any attached files.

Reviewer #1: No

---

## [Author Response · Author response to Decision Letter 0]

24 Nov 2020

Dear Chiara Lazzeri

We thank you to allow us to submit an R1 version of our manuscript.

You will find below our responses to the reviewers comments.

Sincerely 

Raphael Borie and Geréard Zalcman

Journal Requirements:

The R1 manuscript was modified to apply PLOS ONE's style requirements.

2. Please provide additional details regarding participant consent. In the ethics statement in the Methods and online submission information, please clarify whether (1) consent was informed and (2) how oral consent was documented and witnessed. If your study included minors, state whether you obtained consent from parents or guardians. Also, please clarify whether ethics approval and informed consent was obtained from patients in the prospective and retrospective cohorts.

Every patient from the prospective and retrospective cohorts received oral and written information. Please find enclosed the information sheet validated by ethics local committee given to the patients (in French)

Please note that no minor was included in our study.

Every patient has the right to oppose to the use of personal data, though in absence of expressed opposition, we are allowed to use and publish them. As of Nov. 15, not a single opposition was received concerning the current study, the manuscript was modified accordingly

"According to French regulatory laws, all patients whether from the prospective or the retrospective cohorts received written information by the referring physician and provided their oral consent for data collection. All patients were informed about steroids’ and anakinra’s off-label use. Patients’ clinical charts were prospectively collected using a de-identified form. This study was approved by the local ethics committee (CEERB) of Paris Nord (Institutional Review Board-IRB 00006477, Paris-7 University, AP-HP), and by the Institutional Review Board of the French Learned Society for Respiratory Medicine (CEPRO 2020-023). Please find enclosed such approvals. No patient denied to be part of the study at the date of November, 15 2020.

3. In your Methods section, please provide additional information about the participant recruitment method for the prospective part of your study and the demographic details of your participants. Please ensure you have provided sufficient details to replicate the analyses such as:

a statement as to whether your sample can be considered representative of a larger population, and a description of how participants were recruited.

All consecutive inpatients from Bichat Hospital fulfilling the inclusion criteria of the study were accrued, and were representative of all moderate to severe hospitalized COVID-19 patients. During this period of accrual, the CRAs of the study double checked all inclusion/exclusion criteria and eCRF. They also checked that no patient received such drugs, and fulfilled the inclusion criteria and the institutional therapeutic algorithm have been missed, in the prospective collection of cases from the prescription software (see the study flowchart, figure 1). 

All consecutive inpatients from Bicetre Hospital with the same inclusion criteria (level of oxygen supply to ensure a pre-determined level od O2 situation and biological inflammation criteria) of the study were then accrued. All consecutive patients with the inclusion and exclusion criteria were retrieved from the local database. 

Accordingly this cohort is representative of currently available data on moderate to severe COVID-19 patients : majority of men, of elderly people over 65 years, with comorbidities (hypertension, diabetes, moderate overweight, cardiovascular chronic diseases....), all having around 7 days of symptoms evolution, all having bilateral typical pneumonia on CT-scan (performed in all patients), and according to the inclusion criteria, all having elevated inflammation markers along with oxygen needs of 3L/min or more to ensure at least 94% of SaO2.

Eventually as our study confirmed previously reported prognosticators (age, male gender, hypertension, diabetes, active smoking, increased CRP, and elevated D-dimer levels) (see Table 4).

We suggest adding the following sentence in the discussion.

The therapeutic algorithm established and validated by the consortium of MDs in charge of COVID-19 patients in Medicine departments (ICU department were not included since ICU transfer was an endpoint of the study) was distributed to all prescribing doctors (including titular doctors or junior doctors i.e. residents) of the hospital and the information on the prospective study provided to the more than 80 physicians involved. So all patients with the bona fide inclusion criteria were accrued in this observational study

4. For the retrospective data collected in your study, please include the date(s) on which you accessed the records to obtain the data used in your study.

The date of inclusion of such patients from the retrospective was comprised between 28th Feb. to 4th Apr.2020; 85% of these control patients being included (hospitalized) during March. It has been added in the revised MS

5. Please provide the name and catalog number of the RT-PCR kit used.

For all patients included in this study, diagnosis of SARS-CoV-2 infection was performed by RT-PCR on naso-pharyngeal swabs. Different techniques were performed throughout the study period, due to frequent shortage issues and requirements for fast turnaround time: RealStar® SARS-CoV-2 (Altona, Hamburg, Germany), Cobas® SARS-CoV-2 (Roche Diagnostics, Branchburg, NJ, USA), Simplexa® COVID-19 Direct kit (DiaSorin, Gerenzano, Italy), BioFire® SARS-CoV-2 (BioMerieux, Salt Lake City, UT, USA) and QIAstat-Dx® Respiratory SARS-CoV-2 (Qiagen, Hilden, Germany). 

The name and catalog number of the kit is now reported in the manuscript.

6. Thank you for stating the following in the Competing Interests section:

"Raphael Borie, Laurent Savale, Antoine Dossier, Camille Taillé, Benoit Visseaux, Kamel Jebreen, Sébastien Ottaviani, Chloe Tesmoingt, Lise Morer, Tiphaine Goletto, Nathalie Faucher, Linda Hajouji, Catherine Neukirch, Mathilde Phillips, Sandrine Stelianides, Solenn Brosseau, Johan Pluvy, Marie Pierre Debray, Raynaud-Simon Agathe, Antoine Khalil, Vincent Descamps, Thomas Papo, Marc Humbert, Bruno Crestani, Eric Vicaut, Gérard Zalcman have nothing to disclose.

Jean Francois Timsit reported participation to an advisory board from Gilead. Is the principal investigator of PHRC-N 'Covidicus' (Dexamethasone vs. Placebo on Covid-19 pneumonia in ICUs) granted by the French Ministry of Health.

Jade Ghosn reported receiving travel grants and fundings from Gilead Sciences, ViiV Healthcare and MSD.

Benoit Visseaux reported grants from QIAGEN outside the scope of the current work

Xavier Lescure reported travel grants and fundings from Gilead, MSD, Astellas, Eumedica."

Please confirm that this does not alter your adherence to all PLOS ONE policies on sharing data and materials, by including the following statement: "This does not alter our adherence to PLOS ONE policies on sharing data and materials.” 

Patients Data base used for the present analyses will be available on request after approval of the ethical committee of Assistance Publique -Hôpitaux de Paris (AP-HP) data warehouse. Data requests may be sent to Pr G. Zalcman, Service d’Oncologie Thoracique, Hôpital Bichat-Claude Bernard, 46 rue Henri Huchard 75018 Paris, France.

We confirm that such statement does not alter our adherence to PLOS ONE policies on sharing data and materials and we added the suggested statement

Accordingly we updated competing interests statements in the cover letter

7. One of the noted authors is a group or consortium (Bichat & Kremlin-Bicêtre AP-HP COVID teams). In addition to naming the author group, please list the individual authors and affiliations within this group in the acknowledgments section of your manuscript. Please also indicate clearly a lead author for this group along with a contact email address.

We are sorry for this misunderstanding: the group of investigators from both hospitals from the Great Paris University Hospitals (entitled in French "Assistance Publique-Hôptaux de Paris", acronym being AP-HP) is constituted by the investigators listed as authors of the current paper, who actually gathered to a share commune strategy to take care of COVID-19 patients in their respective hospitals. Therefore, there are no additional authors to credit neither any other lead author.

Reviewers' comments:

Reviewer's Responses to Questions

Comments to the Author

1. Is the manuscript technically sound, and do the data support the conclusions?

Reviewer #1: Partly

2. Has the statistical analysis been performed appropriately and rigorously? 

Reviewer #1: Yes

3. Have the authors made all data underlying the findings in their manuscript fully available?

Patients Data base (raw data) used for the present analyses are available on request

. Data requests may be sent to:

- Comité d’évaluation de l’éthique des projets de recherche biomédicale (CEERB Paris Nord) IRB00006477 sec.ceerb@aphp.fr; or michel.lejoyeux@aphp.fr

or

 Pr G. Zalcman, Service d’Oncologie Thoracique, Hôpital Bichat-Claude Bernard, 46 rue Henri Huchard 75018 Paris, France.

We confirm that such statement does not alter our adherence to PLOS ONE policies on sharing data and materials and we added the suggested statement

Reviewer #1: No

4. Is the manuscript presented in an intelligible fashion and written in standard English?

Reviewer #1: Yes

5. Review Comments to the Author

Reviewer #1: The manuscript by Raphael Borie and Colleagues explore an interesting and up to date field regarding the possible immunosuppressive therapies in course of the inflammatory phase of Covid-19, but some concerns needs to be clarified.

Major points

- The corticosteroid group is heterogeneous, including also patients receiving not only steroids, but also anakinra (35/108). The authors assess that anakinra group is too small in order to drive any conclusion about the effectiveness of this drug. In any case it is opinion of the reviewer that some information about this biologic treatment should be shown: at least its effect regard PCR, Ferritin, Fibrinogen, D-dimer concentration, PaO2/FiO2 ratio …. Otherwise anakinra group should be removed.

We thank Reviewer 1 for his/her comment. However we do not share his/her interpretation of the design of our study.

Indeed, our patients received either corticosteroids alone, or corticosteroids plus ankinra, and this is the main reason why it is impossible to individualize the specific contribution of ankinra. 

We have to replace it in the history of COVID-19 pandemic: the study was performed during the first European wave in late March/ early April 2020, when nothing was known about the natural history of the disease. 

This was therefore a pragmatic, real-life, observational study to assess whether our therapeutic standard, collectively adopted by all the COVID-19 physicians, including corticosteroids, while at that time, their use was firmly discouraged by WHO. It was three months before Recovery trial results were released. Not a single study with anakinra was published either at that time.

Our primary endpoint, at that time of a huge afflux of patients, was to avoid death and ICU transfer for mechanical ventilation, while our ICU departments was overwhelmed. 

Concurrently, we wanted to be sure not cause harm to our patients with such a standard of care not supported at that time by data from randomized trials, but only by observational data from our Italian and Chinese colleagues with whom we were in contact. In patients whose condition was straightaway poor or who presented with worsening respiratory conditions despite "standard treatment" and corticosteroids, we totally lacked therapeutic alternatives, and this was the reason why we decided to introduce the anti-IL1 anakinra in our therapeutic algorithm, based on scarce data in other hyper-inflammatory diseases. 

But our aim was not, of course, to compare face to face corticosteroids versus corticosteroids + anakinara since our design was not a based on a randomization, but rather sort of a run-in to ensure at least the lack of detrimental effect. 

Thus, we absolutely did not aim to assess whether or not, anakinra was able to decrease inflammation parameters. Accordingly, in such a pragmatic study, we did not systematically record such parameters evolution at D15, and we cannot provide such a comparative analysis between D1 and D15 biological parameters. 

However, several reports of the use of Anakinra in COVID-19 have been published since then, that actually showed some ability to decrease markers of inflammation (see: Cavalli G et al. lancet Rheumatol 2020 https://doi.org/10.1016/ S2665-9913(20)30127-2). 

Since we completed our study, the efficacy of Anakinra on patients' respiratory condition and survival was not convincingly supported in randomized trials (most of them using ankinra without corticosteroids). Indeed, one French randomized trial (ANACONDA-COVID 19, sponsor = University Hospital of Tours) was actually stopped on Oct. 29th, because of a higher number of deaths in the anakinra arm (press release). According to another press release, another French trial (with an emulated control group), CORIMMUNO, sponsored by AP-HP (Great Paris University Hospitals), also concluded to the absence of any significant effect on survival with anakinra (again without corticosteroids).

Although the primary endpoint of our prospective study was to evaluate a complete therapeutic algorithm relying on high dose corticosteroids (CS), with anakinra as a rescue, we did perform the analysis suggested by Reviewer 1. Again, one has to remind that patients receiving anakinra in these series were those with the most severe respiratory presentation or those in whom CS failed to improve the respiratory failure. 

When the total 35 patients who ever received anakinra (in 1L or 2L) with corticosteroids (one patient only received anakinra but was not analyzed in this subset to stay homogeneous), was analyzed, they were actually more severe than the others, as expected, with a median O2 flow of 8L/mn (5.0-15.0) vs. 5.0 in the whole population, and 5.0L/mn in control patients. 

The COVID-19 related death risk was 10/35 (28.6%) vs. 19/63 (30.2%) in the control group, both groups having also similar rates of death or ICU transfer of 40.0% (14/35) and 39.7% (25/63) respectively. Non-adjusted OR for death was 0.93 for Anakinra group vs. control group (0.37-2.30), p= 0.87, and non-adjusted OR for death or ICU transfer was 1.01 (0.44-2.36), p=0.97.

So actually they did not worse than the control, although presenting with more severe condition, but taking into account the negative results of randomized trials, and the low sample size of the Ankinra group, we feel it would be very imprudent to include such a subset, unplanned analysis

- Moreover the authors should report any possible side effect, or absence of side effects of anakinra treatment compared to steroids alone.

We mentioned, page 16 all adverse effects that we observed, with at least 15 days of follow-up, mainly linked to corticosteroids use. We did not observe specific adverse effects that we could have been imputed to anakinra, but again, patients who received anakinra in our series were also treated by corticosteroids, and it is the main difference with all published papers on the use of Ankinra in severe COVID-19 to date. Of note the infections complications (one patient who presented with bacterial pneumonia and aspergillosis and one with varicella-zoster), both occurred in patients receiving only corticosteroids, none occurring in the anakinra group, with a web-based follow-up until 30 days after discharge:

"Short-term toxicity within the first 15 days was not unexpected, essentially related to pre-existing type 2 diabetes exacerbations in 29 of 108 patients (26.9%), with oral anti-diabetic drugs alone required in 10 patients (34.5%), and insulin therapy in 19 patients (65.5%). Only three patients with steroid-induced diabetes exacerbation died, all from respiratory distress, while diabetes was correctly controlled with insulin. Pseudomonas aeruginosa pneumonia and invasive aspergillosis occurred simultaneously in one patient following ICU admission. A case of abdominal varicella-zoster occurred in one patient, with favorable evolution upon valaciclovir."

One has to remember that we used lower doses of ankinra as compared with other studies (many used 100 mg IV/SC every 6 hrs, not every 24 hrs), since it was the dose evaluated in large phase 3 trial assessing ankinra in sepsis, and since it was associated with corticosteroids in our study: we actually wanted to reduce the risk of infectious adverse events.... and we succeeded not harm patients. 

Indeed, in his series, Cavalli reported 14% of bacteriema with isolation of Staphylococcus epidermidis in their patients, with such higher dosed ankinra, but also in the 7 patients with a more standard dose of sub-cutaneous 200 mg daily anakinra.

- According to the inclusion criteria for anakinra therapy, only patients requiring ≥ 6L /min O2 therapy from day one, or ≥ 3L /min O2 therapy from day 4, received anakinra (together with steroids), while those patients that were in better clinical condition received steroids alone. It is difficult to expect a better outcome for patients starting from a worst clinical condition, independently from the therapy regimen. If no differences raised from the comparison of the two groups, then the author can drive the conclusion that anakinra does not get worst the prognosis. This possible result implicitly suggests some possible positive effect of anakinra, being the starting population in a more severe condition. In any case, as the authors assess, a controlled randomised study is needed.

We perfectly agree with Reviewer's feeling although such assertion is currently impossible to prove because of our study design. Moreover, the negativity of randomized trials using ankinra since we completed our study (see above) does not support such hypothesis, unless, actually the combination of corticosteroids and ankinra in our study could have potentiated the putative effect of anakinra in the most severe patients.

Again, the aim of the study was not to compare the to groups of patients, those only receiving corticosteroids and those receiving combination treatment because of higher severity but, as mentioned above, to appreciate the impact of the strategy in the whole cohort.

- What are the result of the comparison between the control group (n 63) respect to steroids only group (n 70)?

Owing to the smaller sample size of such unplanned subset analysis, the OR for death of corticosteroids-only group vs. control group was 0.48 (0.21-1.09), p=0.08 and the OR for death/ICU transfer corticosteroids-only group vs. control group was 0.54 (0.25-1.10), p=0.087, supporting the role of corticosteroids, since their beneficial effect is still obvious. 

Again as the primary aim of the study was the evaluation of the therapeutic algorithm as a whole, and we feel it would be methodologically unsuitable to provide such unplanned analysis, but are ready to do it, if the Editor and the Reviewer insist.

- The control group received less frequently standard therapy as compared to corticosteroid ± anakinra group. This could be a bias of the study, even if the probability that some of these drugs could significantly modify the natural history of the disease is low, as the authors assert in the discussion. The statistical analysis should be reconsidered after removing, from the control group, those patients not treated with the same standard therapy as the corticosteroid ± anakinra group.

In that study form the early period of the COVID-19 first wave, what was called "standard therapy" was oxygen supply of course, systematic thrombophylaxis by LMWH, , the antiviral lopinavir/ritonavir and the anti-parasite Ivermectin (both drugs subsequently shown of no efficacy on SARS-CoV-2), and the combination of hydroxychloroquine and azithromycin, the latter unfortunately based on media noise on such association, rather than scientific evidence, unfortunately (see sTable 1). 

A sensitivity analysis further showed that the use of Tociluzimab in 3 patients (instead of Ankinra, protocol deviation) and of Remdesivir (instead of lopinavir/ritonavir) in 3 otheer patients, did not change the results when such patients were excluded form the analysis.

The difference between the control group and the patients treated with corticosteroids actually only derived from the fact that less control patients receive HCQ, azithromycin, Lopinavir/ritonavir or ivermectin as shown on Table s1, again all these drugs being proved to be inefficient subsequently.

We currently know from several randomized large trials published during last 4 months, that HCQ+azithromycin has virtually no beneficial effect on survival, and conversely could have induced more deaths by cardiac arrhythmia, as reported by NIH and FDA on the basis of large published evidence. 

We therefore think that an additional subset analysis with a lower sample size would not be adequate, and would have no chance to change our results, since our series treated with corticosteroids +/- ankinra potentially received a survival disadvantage by receiving not only inefficient but also potentially deleterious drugs, while they hopefully received corticosteroids, the only efficient drug so far in severe COVID-19.

Minor points

- Abstract, line 12: it is not clear if the authors refer to the “corticosteroids + anakinra group” or to the “corticosteroids + anakinra, plus the corticosteroids alone group”. The sentence “corticosteroids ± anakinra group” should be used.

Thank you for such remark: we actually corrected the sentence as suggested

- Discussion, page 13, line 13: “anakinra or steroids” should be probably changed with “corticosteroid ± anakinra”

We did change the sentence as suggested

---

## [Editor Report · Decision Letter 1]

2 Dec 2020

Glucocorticoids with low-dose Anti-IL1 Anakinra Rescue in Severe Non-ICU COVID-19 Infection: a Cohort Study

PONE-D-20-25884R1

Dear Dr. Borie,

We’re pleased to inform you that your manuscript has been judged scientifically suitable for publication and will be formally accepted for publication once it meets all outstanding technical requirements.

Kind regards,

Chiara Lazzeri

Academic Editor

PLOS ONE
---

## [Editor Report · Acceptance letter]

7 Dec 2020

PONE-D-20-25884R1 

Glucocorticoids with low-dose Anti-IL1 Anakinra Rescue in Severe Non-ICU COVID-19 Infection: a Cohort Study 

Dear Dr. Borie:

I'm pleased to inform you that your manuscript has been deemed suitable for publication in PLOS ONE. Congratulations! Your manuscript is now with our production department. 

Kind regards, 

on behalf of

Dr. Chiara Lazzeri 

Academic Editor

PLOS ONE